# Development of SNP Markers from GWAS for Selecting Seed Coat and Aleurone Layers in Brown Rice (*Oryza sativa* L.)

**DOI:** 10.3390/genes13101805

**Published:** 2022-10-06

**Authors:** Me-Sun Kim, Seo-Rin Ko, Van Trang Le, Moo-Gun Jee, Yu Jin Jung, Kwon-Kyoo Kang, Yong-Gu Cho

**Affiliations:** 1Department of Crop Science, College of Agriculture and Life & Environment Sciences, Chungbuk National University, Cheongju 28644, Korea; 2Division of Horticultural Biotechnology, Hankyong National University, Anseong 17579, Korea

**Keywords:** brown rice, seed coat, aleurone layer, genome-wide association study (GWAS), core collection, HRM markers

## Abstract

Ninety-five percent of the general nutrients in rice are concentrated in the rice bran and germ, and many nutrients such as vitamins, minerals, dietary fiber, and essential fatty acids, as well as antioxidants such as tocopherol, are lost during milling. In this study, we investigated the thickness of seed coat and aleurone layers using a 294 rice core collection, and found candidate genes related to thickness of seed coat and aleurone layers, by performing a genome wide association study (GWAS) analysis using whole genome resequencing data. Two primer pairs that can be used as high-resolution melting (HRM) markers were developed. As a result of genotyping BC_2_F_2_ individuals derived from a cross between “Samgwang” and “Seolgaeng”, and using corresponding HRM markers, it was possible to finally develop HRM markers for selecting seed coat and aleurone layer thickness. This is expected to be used as basic data for the application of gene editing using CRISPR/Cas9 technology and for establishing a breeding strategy for high eating quality rice using molecular genetic technology.

## 1. Introduction

Rice is one of the major grains in the world, especially in Asian food culture, and is used as a staple food [1]. In general, the main component of rice is starch, which accounts for more than 70%, and it also contains 7–8% protein, 1–3% fat, and trace vitamins and minerals [2]. Rice contains more than 95% of its general nutrients in the rice bran and germ [3]. As such, many nutrients such as vitamins, minerals, dietary fiber, and essential fatty acids and antioxidants such as tocopherol are lost during milling [4,5]. Brown rice is composed of an embryo, germ, and rice bran layer, and is richer in minerals such as calcium, phosphorus, iron, vitamins B1, B2, B6, nicotinic acid, pantothenic acid, and folic acid than white rice [6]. In addition, brown rice contains twice as much dietary fiber as white rice, so it acts as a major source of dietary fiber.

Genome-wide association study (GWAS) is a method of searching the entire genome for phenotypes showing continuous variation and mutations representing associations in natural individuals, or for lines using whole genome sequences [7,8]. In order for GWAS to be performed, a minimum of 30,000 to a maximum of 1,000,000 SNPs are required, and it is possible to identify agriculturally important trait-related genes by performing the recombination association analysis that occurs naturally in the evolutionary process [9,10,11,12]. Many GWAS studies have been reported on the discovery of genetic mutations causing large phenotypic variation using such large-scale SNP marker information [13,14,15,16]. The mutations reported so far in GWAS analysis have been based on Nipponbare, a japonica cultivar as a standard variety, but bioinformatics department of IRRI has additionally discovered various genetic mutations by creating genetic mutation maps using whole genome sequencing, such as Kasalath and IR64 [17,18,19].

A core collection should be a group that can be used to analyze multiple traits, including as much genetic diversity as possible, with the minimum number of genetic resources in one crop, and should have high usability in research. The reduced size of a core collection is the key to manageability and represents the diversity of the entire collection, allowing analysis to be handled more efficiently and effectively. Zhao et al. [10] performed genotyping on 413 rice genetic resources collected from 82 countries worldwide, to discover 44,100 SNP mutations, and performed GWAS analysis using them. The GWAS analysis on five rice traits was carried out using a total of 315 rice cultivars from the International Core Rice Germplasm Bank, detecting a total of 36 candidate genes [20]. In addition, the USDA (United States Department of Agriculture) rice mini-core collection (URMC) has been successfully used by researchers to identify sources of germplasm and new QTLs for several important characteristics, such as grain quality, starch quality, grain yield, abiotic, and biotic stress tolerance [21,22,23,24,25].

In this study, we investigated the thickness of seed coat and aleurone layers using a 294 rice core collection, and candidate genes related to thickness of seed coat and aleurone layers were found by performing GWAS using SNPs in whole genome resequencing. In this way, SNP markers were developed to provide efficient molecular tools for improving the eating quality and processability of brown rice in breeding programs. This study will contribute to the understanding of the mechanisms of seed coat and aleurone layer formation in rice and will provide a basis for future research for breeding brown rice with an excellent taste and processability.

## 2. Materials and Methods

### 2.1. Plant Materials

Regarding the rice genetic resources used in this study, 166 core groups were first selected using introduced species and domestic breeding lines, weedy rice, and IRRI resources among 24,000 genetic resources collected from around the world. A total of 294 rice genome resequencing groups were finally constructed by secondarily selecting additional varieties that can be headed in the domestic climate and have a high utility value in Korea. Of the total 294 rice genome resequencing groups, the 225 domestic breeding lines consisted of 38 Indica, 178 Japonica, and 9 Tropical Japonica [26].

### 2.2. Phenotypic Analyses Using Oil Red O Staining

Seeds of 294 brown rice genetic resources were cut into two halves using a razor blade, and then seeds were stained with Oil Red O dye used for lipid staining [27]. Oil Red O stock solution was prepared by dissolving 0.5 g of Oil Red O (Sigma, St. Louis, MI, USA) in 100 mL of isopropanol. Oil Red O working solution was prepared by mixing 6 mL of Oil Red O stock solution and 4 mL of distilled water. Half-cut seeds were stained in Oil Red O working solution for 10 min and then washed 3 times with 70% ethanol. The structure and thickness of the seed coat and aleurone layers were measured using an Eclipce E600 microscope (Nikon, Tokyo, Japan). For images of half-cut staining seeds, the total cutting area and staining area were calculated using the ImageJ program, to measure the thickness of the seed coat and aleurone layers [28,29]. The experiment was repeated at least 4 times and the results are expressed as average values [30]. 

### 2.3. DNA Isolation and Genome Sequencing

Genomic DNA of 294 rice accessions was extracted using DNeasy Plant Mini Kits (QIAGEN, Hilden, Germany), according to the kit manual instructions from 0.5 g of young leaf tissues. Quality of DNA was checked by diluting a sample extracted using a Quant-iT BR assay kit (Q32850, Invitrogen) and then measuring the concentration and quality of DNA. Tecan F200 (Tecan, Männedorf, Switzerland) was used to measure the optical density value, and the extracted DNA was confirmed by electrophoresis at 1% agarose gel. Genome re-sequencing was performed on an Illumina HiSeq 2500 platform (Illumina, Inc., San Diego, CA, USA), according to its standard sequencing protocol. Data analysis was compared with IRSGP1.0 downloaded from NCBI (https://www.ncbi.nlm.nih.gov/assembly/GCF_001433935.1/ accessed on 30 July 2019), which is a standard genome sequence of rice [26]. Raw data files from short pair-end sequencing reads were first filtered using the software Cutadapt and Sickle, respectively [31,32], and then mapped to the reference genome using BWA software [33]. Removal of duplicate reads and SNP calling were analyzed using GATK software [34], and high-quality SNPs were filtered based on <10% of missing data and >5% of minor allele frequency. After filtering, a total of 1,842,515 SNPs were used for GWAS analysis.

### 2.4. GWAS

GWAS was performed using a genomic association and prediction integrated tool, GAPIT developed by Cornell University. GAPIT is a program that can be applied to the quantitative characteristics of plants, such as flowering time and seed weight. A mixed linear model of GAPIT was used in this study [35]. Significant SNPs were determined based on the threshold of −log *p* > 5.

### 2.5. Analysis of the LD Block and Haplotype

We used Haploview (v.4.2) for analysis of the LD block and haplotype analysis [36]. We defined LD blocks according to definition of haplotype block described by Gabriel et al. [37]. This method defines pairs as in strong LD if the one-sided upper 95% confidence bound on D′ is >0.98 and the lower bound is above 0.7. Each color represents the magnitude and significance of pairs LD, and the gradation from red to white is from a high LD value to a low value. Red diamonds without numbers correspond to a D’ value of 1.0. LD analysis was used to designate a region containing 10 kb either side of the lead SNP as the candidate region. Annotations of genes within candidate regions were derived from the Rice Annotation Project database (https://rapdb.dna.affrc.go.jp accessed on 30 July 2019). Haplotypes were used for comparative phenotypic analysis.

### 2.6. Identification of SNPs and Candidate Genes

Adjacent significant SNPs associated with target traits within a physical distance of 200 kb were screened for candidate genes. Genomic DNA sequence and cDNA region for the mRNA transcripts of each candidate genes were searched and extracted from the Rice Genome Annotation Project (https://shigen.nig.ac.jp/rice/oryzabase/ accessed on 30 July 2019) and the NCBI database (https://www.ncbi.nlm.nih.gov/ accessed on 30 July 2019) [38].

### 2.7. Development of Molecular Markers 

Molecular markers associated with the seed coat and aleurone layers were developed for high resolution melting (HRM) analysis. HRM specific primer sets based on target SNPs were designed using the web-based software Primer3plus (https://primer3plus.com accessed on 23 September 2020) [39]. HRM analysis was conducted using the ECO48 Real-time PCR System (PCRmax, Staffordshire, UK). A total of 20 μL of reaction mixture contained 40 ng of DNA template, 10 μM of each primer, and 10 μL of 2X qPCBIO HRM mix (PCR Biosystems Ltd., London, UK), including a SybrGreen fluorescent dye. PCR amplification was conducted with the following conditions: 95 °C for 5 min; 40 cycles of 95 °C for 15 s, 60 °C for 15 s, and 72 °C for 30 s. For HRM analysis, PCR amplicons were denatured at 95 °C for 15 s and then held at 55 °C for 15 s, to anneal the DNA duplexes. These steps were followed by a melting curve ranging from 65 to 95 °C, with temperate increments of 4.4 °C per second. The fluorescence data were processed to generate melting curves for SNP calling using the ECO48 Software (PCRmax, Staffordshire, UK).

## 3. Results

### 3.1. Phenotypic Variation for Traits Related to Seed Coat and Aleurone Layers

For the accuracy of seed coat and aleurone layers thickness, the cross-section of each seed was divided into five parts, namely the dorsal, upper lateral, lateral, lower lateral, and ventral part. The distribution of the seed coat and aleurone layer thickness of 294 rice core collection is as shown in Figure 1. The thickest part and the middle parts were mostly concentrated around the dorsal and upper lateral parts. The thinnest part was found to be widely dispersed in the seed section, but more concentrated around the ventral part. Most of the evaluated traits showed a normal distribution (Figure 2). Maximum, minimum, mean, standard deviation (SD), and coefficient of variation (CV) values are provided in Table 1. As a result of investigating the distribution by section based on the thickest aleurone layers region, the highest distribution was found in section of 80 px < x ≤ 100 px, and “Heugseol”, which has temperate japonica ecotype, was 278.26 px, showing the greatest aleurone layer thickness among the 294 rice core collection. In addition, in the distribution by section based on the thinnest aleurone layers region, the highest distribution was shown in the section of 10 px < x ≤ 15 px, and “Tongil” of indica type was 4.20 px, displaying the thinnest aleurone layers. The CV values of the 294 rice resources ranged from 31 to 40%, suggesting a considerable association with a wide phenotypic distribution. 

### 3.2. Genome-Wide SNPs and Association Analysis

Quality control was performed, to select criteria for selection of specific SNPs related to seed coat and aleurone layers thickness for re-sequencing the data of the 294 rice core collection. Of the total of 6,243,699 SNPs, 1,842,515 (29.5%) SNPs with a read depth of 4× or more and 200× or less, MAF > 0.05, genotyping quality ≥ 20, and Hardy–Weinberg equilibrium (*p* ≤ 0.001) were confirmed. PCA and kinship analyses were performed for population structure analysis using high-quality 1,842,515 SNPs, with a compressed mixed linear model (CMLM). A 294 rice core collection panel was determined by graphical representation of the K value on a screen plot, which indicated the presence of three sub-populations. This result was confirmed by the variation in the first 10 major components (PCs) showing inflection points in PC2, suggesting that the first two PCs dominated the population structure on the association analysis (Figure 3). The first two PCs were significant and explained 39.65 and 3.7% of the total genetic variation, which suggests the predominance of these three components in controlling the population structure of association panel. A kinship matrix is displayed as a heatmap, where red indicates the highest correlation between pairs of individuals and yellow indicates the lowest correlation. The relatedness of pairwise coefficients estimated in the kinship matrix indicated a lower genetic relatedness among individuals in the association panel. 

GWAS analysis using GAPIT usually uses about 1 million SNPs to simultaneously test for associations with traits [35]. GWAS was performed using phenotypic data on the seed coat and aleurone layer thickness of 294 rice resources. Through analysis of the Manhattan plot results of GWAS, SNPs showing a correlation with seed coat and aleurone layers thickness were discovered for chromosomes in each group. The Manhattan plot and QQ plot are shown in Figure 4 based on the −log_10_(*p*) values of significant SNPs from the correlation analysis of the seed coat and aleurone layers thickness. It was confirmed that significant SNPs were detected based on −log_10_(*p*) > 5 in the Manhattan plot, according to the thickness of seed coat and aleurone layers. It was found that SNPs located on rice chromosomes 1, 3, 4, and 10 were more significant than other SNPs. In addition, the expected *p*-value of *x*-axis and observed *p*-value from significant SNPs showed a diagonal linear shape in the QQ plot, which means that the discovered SNPs expressed their characteristics well, with normality and significance. As a result of searching for SNPs that were correlated with aleurone layers thickness in this study, the top 10 most significant SNPs by region of aleurone layers are shown in Table 2. One significant SNP associated with the thickest aleurone layers part was ‘S4_34033252’, with a −log_10_(*p*) value of 5.94. The SNP related to the middle aleurone layers part was ‘S3_20312675’, which was highly associated with a –log_10_(*p*) value of 6.53. In addition, a significant SNP related to the thinnest aleurone layers part was ‘S1_6459856’, which showed a −log10 value of 6.00. The available haplotype analysis formats are ped and map files; the ped file requires individual information and individual genotype data for each marker, and the map file requires a marker name and marker location. LD is displayed as pairwise D’ values. Each color represents the magnitude and significance of pairs of LD, and the gradation from red to white is from a high LD value to a low value. Red diamonds without numbers correspond to a D’ value of 1.0. Figure 5 shows the results of LD block analysis on chromosomes 1, 3, and 4, respectively, based on SNP, with the highest correlation according to the thickness of the seed coat and aleurone layers. A total of 10 LD blocks were observed on chromosome 1, and a total of five LD blocks were observed on chromosome 3. Six LD block was observed on chromosome 4, and block 4 showed the largest size at 150 kb. Among them, S1_6459856 was determined to be a significant SNP, and candidate genes were searched using Ensemble plant and NCBI, centering on the discovered SNPs. Except for genes whose function was not revealed, ROS1, which was expressed for the thickness of aleurone layers, was selected as the closest candidate gene.

### 3.3. Identification of Candidate Genes Related to the Thickness of the Seed Coat and Aleurone Layers

In order to find candidate genes affecting seed coat and aleurone layers thickness, candidate genes were identified within the range of 200 kb, using the RAP database (https://rapdb.dna.affrc.go.jp/index.html accessed on 30 July 2019) and NCBI database (https://www.ncbi.nlm.nih.gov/gene/ accessed on 30 July 2019), based on the most significant SNP positions on each chromosome (Table 3, Appendix A). A total of 61 genes were identified on rice chromosomes 1, 3, 4, 5, 7, and 10, while the detailed information about several genes was not reported and annotated as conserved hypothetical proteins. Except for the conserved hypothetical protein, genes related to the floral signal such as FLOWERING LOCUS T (FT)-Like homolog, genes related to plant growth such as ORIGIN RECOGNITION COMPLEX 3, and genes related to stress resistance such as CORONATINE INSENSITIVE1 were identified. In addition, genes related to seed development, such as CRINKLY4 and REPRESSOR OF SILENCING 1a, were also identified in strong LD. 

Among them, Os01g0218032 was annotated as repressor of silencing 1a (ROS1a), THICK ALEURONE 2, which displayed functions as DNA demethylase and is related to endosperm development [40]. Os01g0218032 has a gene structure as shown in Figure 6 and is 6.5 kb in length and consists of 17 exons. As a result of identifying allelic mutations of the Os01g0218032 gene in the 294 rice core collection, two missense mutations were found in the first exon of the Os01g0218032 gene. Of the total resources, 74 resources had SNP mutations at the location C6444737T, resulting in mutations from serine to phenylalanine at amino acid location 163. In addition, 76 of the total resources produced SNP mutations at location A6445148C, resulting in mutations from lysine to threonine at amino acid location 301. In the Exon1 region of OsROS1, a total of six SNPs close to S1_6459856 formed LD blocks in the same block as S1_6459856. This is a result of revealing that there is an LD relationship between SNPs. Among the SNP markers in which the LD block was formed, the haplotype for each individual was estimated using Haploview for S1_6444737 and S1_6445148 (Figure 7). For haplotype analysis, the Nipponbare nucleotide sequence was used as standard. The haplotype analysis results and phenotypic data were compared and analyzed, to determine whether there was a relationship between the thickness of seed coat and aleurone layers and the candidate gene. As a result of the haplotype analysis with Os01g0218032, on the basis of the high peak position of rice chromosome 1, the Os01g0218032 gene was divided into two groups. The average thickness of each group was 103.41/73.53 in the thickest part, 50.77/40.04 in the middle part, and 16.12/16.34 in the thinnest part. There was a significant difference among the three genotypes in their thicknesses (with averages of 103.41, 50.77, and 16.12 px) of rice seed coat and aleurone layers. The Os01g0218032-Hap1 group, showing a thick seed coat and aleurone layer, had a base similar to Nipponbare in the exon region, whereas the Os01g0218032-Hap2 group, showing a thin seed coat and aleurone layer thickness, had a base substitution at the exon region.

### 3.4. Development of Molecular Markers for Traits of Seed Coat and Aleurone Layer Thickness

HRM primers were prepared for the regions ‘S1_6444737′ and ‘S1_6445148′ that induce amino acid mutations, to search for associations between seed coat and aleurone layer thickness. Forward and reverse primers containing one putative SNP site were designed for HRM analysis using Primer 3 [39]. The primer pair was designed to have an annealing temperature at 60 ± 1 °C and the product size was expected to be 175–225 bp (Table 4). Among the 294 rice genetic resources, three resources with thin seed coat and aleurone layers, and three resources with thick ones, were selected, and their melting curve patterns were compared using the designed HRM marker. With the HRM results, “Magnolia” (23.21 px), “Hsiang-ha-tsan” (29.55 px), and “Kagi” (33.00 px) were analyzed as the resources with a thin seed and aleurone layer thicknesses, while “Kwanak” (205.12 px), “Jejubukjeju-2002-561” (206.12 px), and “Heugseol” (278.26 px) were selected for resources as thick. As a result of the HRM analysis of each selected resource, according to the thickness of seed coat and aleurone layers, it was confirmed that the melting curve patterns were different, so that they could be used as markers for selecting the thickness of the seed coat and aleurone layers of brown rice (Figure 8A,B). A total of 52 genomic DNA samples were used to analyze each set of primers for HRM analysis, including “Samgwang”, a variety with a thick seed coat and aleurone layers; “Seolgaeng”, a variety with a thin seed coat and aleurone layers; and 50 BC_2_F_2_ (Figure 8C,D). As a result of the HRM analysis using the “TA2S1-737” marker, out of a total of 50 Samgwang X Seolgaeng BC_2_F_2_ groups, 12 individuals with a homozygous genotype (C/C), such as Samgwang, and eight individuals with a homozygous genotype (T/T), such as Seolgaeng, and 30 individuals with a heterozygous genotype (T/C) were identified. In addition, as a result of HRM analysis using the “TA2S1-148” marker, 26 individuals with a heterozygous genotype (C/A), out of 50 BC_2_F_2_ populations of Samgwang X Seolgaeng, were found to match their genotype and phenotype (Table 5).

## 4. Discussion

In rice grains, the embryo and aleurone layers are the major tissues in which lipids are accumulated [27], while the lipid component of brown rice is not only an essential energy source for plant germination and growth, but also an important nutrient source for humans. However, when polished, most of these useful substances are removed, and white rice has the disadvantage of losing food-related ingredients, such as umami-related amino acids, when compared to the unpolished group. 

In this study, we investigated the thickness of seed coat and aleurone layers using a 294 rice core collection, and found candidate genes related to the thickness of seed coat and aleurone layers by performing a GWAS analysis using whole genome resequencing data. SNP markers were developed to select breeding lines for thickness of seed coat and aleurone layers. The cross-section of seeds was measured, with consideration of the accuracy of GWAS for the thickness of the seed coat and aleurone layers. The thickness was measured by dividing the measurement region into five parts: dorsal, upper lateral, lateral, lower lateral, ventral part. Regarding the results of the thickness distribution for the core rice group, the thickest part and the middle part were mostly concentrated in the dorsal and upper lateral parts. CV values ranged from 31% to 40%, suggesting a significant association with a broad phenotypic distribution. We observed large changes in thickness of the aleurone layer and aleurone layer regions among the 294 rice resources. The *japonica* type had a larger staining region than the other types, and most of the *indica* types had a thinner staining region. Our observations are consistent with findings of Khin et al. [27], who reported that the *japonica* type had more aleurone layer cells than the *indica* type. Furthermore, our results showed that the thickness of aleurone layers was dependent on the measurement of regions of the seed, and that the thickness varied significantly between varieties. We found that the seed coat and aleurone layers were the thickest in dorsal region of all varieties. Wang et al. [41] reported that more aleurone cell layers were formed along surface layers of dorsal endosperm, while other surface layers of endosperm formed only a single aleurone cell layer.

According to the genome resequencing results of the 294 rice core collection, the number of sequence reads was about 42 million, and the mapping rate was over 96%. The average coverage was a 9.8 depth, and the total number of SNPs was 11,632,676. In the core collection, the number of SNPs for 30 landrace species was 7,239,907, and the number of SNPs was 6,792,796 for 30 weedy rice. The number of SNPs was 10,728,014, with 65 introduced lines. These results support the fact that the core collection used in this study is suitable for GWAS. 

Brach et al. [42] performed a GWAS analysis using high-quality re-sequencing data for 10 eating and cooking quality (ECQs), in a core collection consisting of 227 non-glutinous genetic resources and four derived resources. In addition, diverse resources, consisting of 751 rice accessions from the 3000 Rice Genomes Project [43], were analyzed through GWAS, to dissect the genetic basis of GNC and GCC mutations, which are important determinants of grain yield and quality [44]. In this study, GWAS was performed using high-quality resequencing data from a 294 rice core collection and phenotypic data, and using a staining method with verified significance. It was confirmed that significant SNPs were detected based on −log10(p) > 5 in a Manhattan plot, according to the thickness of the seed coat and aleurone layers. In addition, the expected *p*-value of the *x*-axis and observed *p*-value from significant SNPs showed a diagonal linear shape in the QQ plot, which means that the discovered SNPs expressed their characteristics well, with normality and significance. In the results of the LD block analysis, based on the SNP with the highest correlation according to thickness of seed coat and aleurone layers, various LD blocks were observed on chromosome 1, 3, and 4; among them, S1_6459856 was determined to be a significant SNP, and ROS1 was selected as the closest candidate gene. As a result of the haplotype analysis with Os01g0218032, on the basis of the high peak position of rice chromosome 1, the Os01g0218032 gene was divided into two groups. The Os01g0218032-Hap1 group, showing a thick seed coat and aleurone layers, had a base similar to Nipponbare in the exon region, whereas the Os01g0218032-Hap2 group, showing a thin seed coat and aleurone layer thickness, had a base substitution at the exon region. 

In this study, as a result of searching for SNPs correlated with seed coat and aleurone layer thickness, it was confirmed that the SNP (S1_6459856) located in rice chromosome 1 was similarly located to ROS1a (Os01g0218032), which is related to seed coat and aleurone layer thickness. Base changes in the OsROS1 gene cause mutations in the aleurone cell layer through inhibition of two putative transcription factor genes, RISBZ1 and RPBF, which increase aleurone layers due to lower levels of DNA hyper-methylation and expression [40]. In null mutants of OsROS1a using CRISPR/Cas9 technology, 378 differentially alternative splicing (AS) genes were identified, indicating that the OsROS1a gene was involved in the aleurone layer thickness and seed development of rice, through the expression and AS regulation of genes associated with endosperm development [45].

As a result of identifying the allelic mutations of the Os01g0218032 gene in the 294 rice core collection, two missense mutations were found in the first exon of the Os01g0218032 gene. S1_6444737 (C > T) is mutated from serine to phenylalanine at amino acid position 163. In addition, in the case of S1_6445148 (A > C), a SNP mutation was made, resulting in a lysine to threonine mutation at amino acid position 301. Three varieties with a thin seed coat and aleurone layers, and three varieties with thick ones, were selected to test genotypes, and their melting curve patterns were compared using the designed HRM markers. It was confirmed that the melting curve patterns were differentiated by genotypes, homo (P1): hetero: homo (P2) (1:2:1), so that they could be used as markers for selecting the thickness of the seed coat and aleurone layers of brown rice. In this study, two primer pairs that can be used as HRM markers were finally developed by GWAS analysis of the 294 rice core collection. As a result of genotyping with the 50 BC2F2 individuals derived from the cross between “Samgwang” and “Seolgaeng”, it was confirmed that different melting curve patterns are shown, depending on the thickness of seed coat and aleurone layers using a corresponding HRM marker. In conclusion, it was possible to develop the HRM markers for selecting seed coat and aleurone layer thickness. SNP-based HRM markers have been studied in various fields, such as disease resistance and susceptibility discrimination, with sequencing through cross breeding. Furthermore, markers for quantitative traits were developed. The markers developed in this study enable efficient testing of cross populations and provide materials that can be applied, not only to aleurone layers, but also in constructing breeding populations using various traits. In addition, gene mapping and QTL analysis are possible using the developed HRM molecular markers.

This is expected to be used as basic data for the application of gene editing technology using CRISPR/Cas9 and for the establishment of a breeding strategy for high eating quality rice using molecular genetic technology.

## Figures and Tables

**Figure 1 genes-13-01805-f001:**
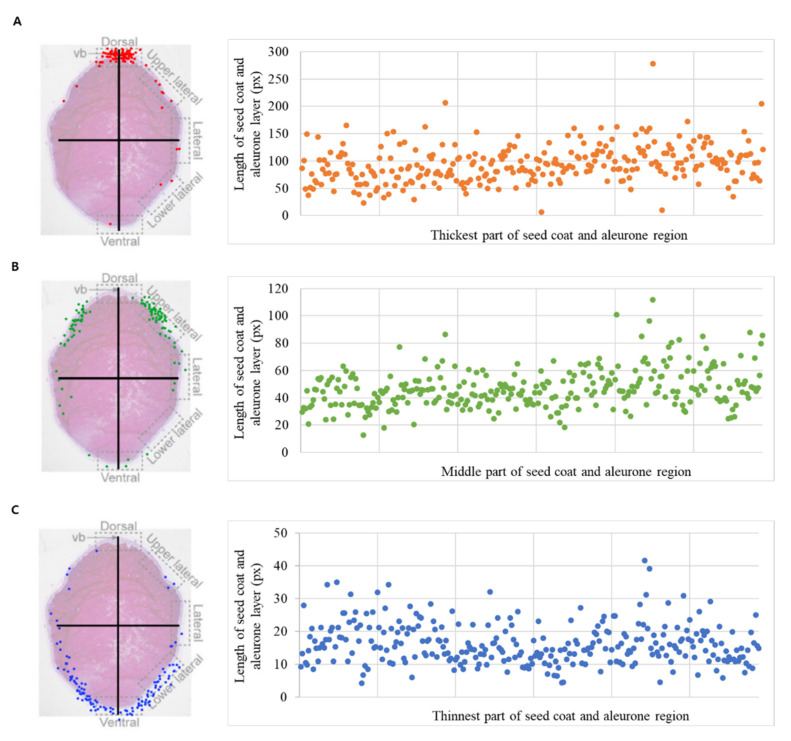
Scatterplot of seed coat and aleurone layer thickness by region, in the 294 rice resources. (**A**) the thickest part, (**B**) middle part, and (**C**) the thinnest part. The cross-section of seeds was classified into a total of five areas (dorsal, upper lateral, lateral, lower lateral, and ventral) according to half-cut seed method and investigated.

**Figure 2 genes-13-01805-f002:**
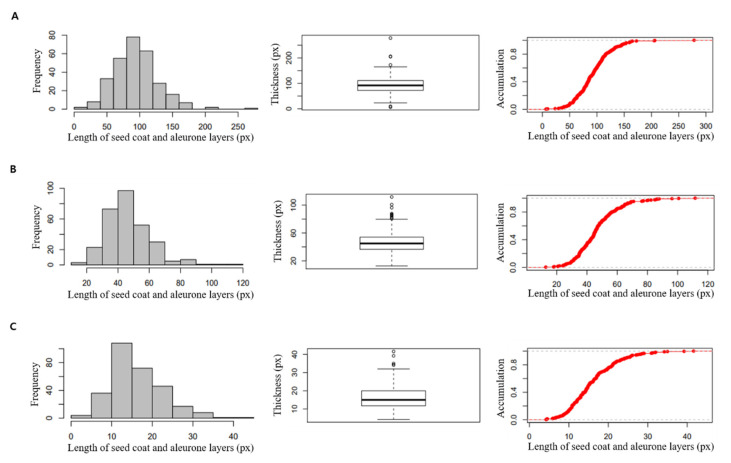
Phenotypic distributions and box plots of seed coat and aleurone layer thickness by region with 294 rice resources. (**A**) the thickest part, (**B**) middle part, and (**C**) the thinnest part.

**Figure 3 genes-13-01805-f003:**
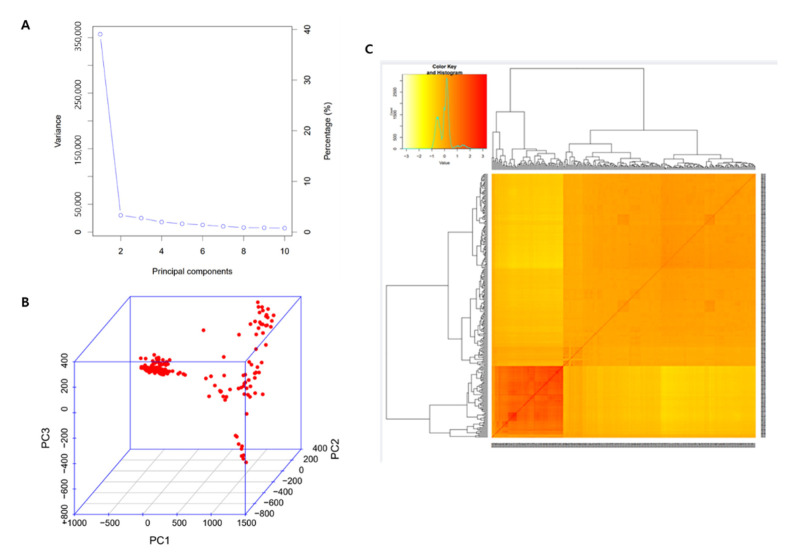
Population structure analysis of the 294 rice core collection. (**A**) Variance of the first 10 principal components reflected by 1,842,515 SNPs in GWAS, (**B**) principal component analysis (PCA) of SNPs, indicating the population structure of the core collection, (**C**) kinship plot showing the relationship among the genotypes.

**Figure 4 genes-13-01805-f004:**
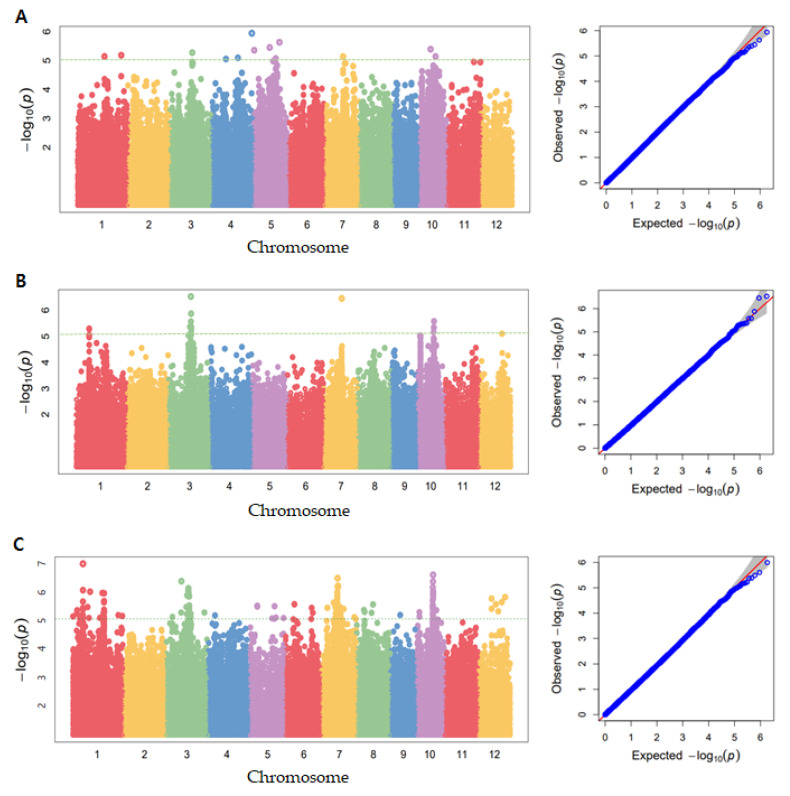
Genome-wide association study of seed coat and aleurone layers by region in the 294 rice core collection. The −log_10_(*p*) values from a genome-wide scan are plotted against the position on each of the 12 chromosomes. (**A**) The thickest part, (**B**) middle part, and (**C**) thinnest part.

**Figure 5 genes-13-01805-f005:**
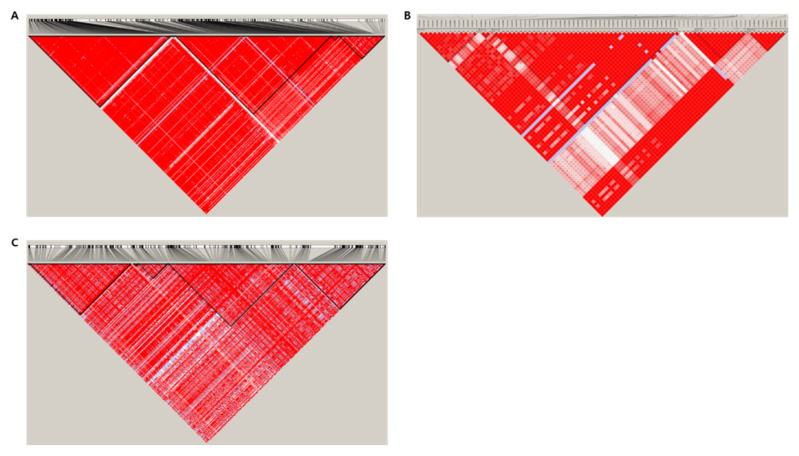
Linkage disequilibrium (LD) plot generated using Haploview software. LD is displayed as pairwise D’ values. Each color represents the magnitude and significance of pairs LD, and the gradation from red to white is from a high LD value to a low value. Red diamonds without numbers correspond to a D’ value of 1.0. (**A**) S4_34033252, (**B**) S3_20312675, (**C**) S1_6459856.

**Figure 6 genes-13-01805-f006:**
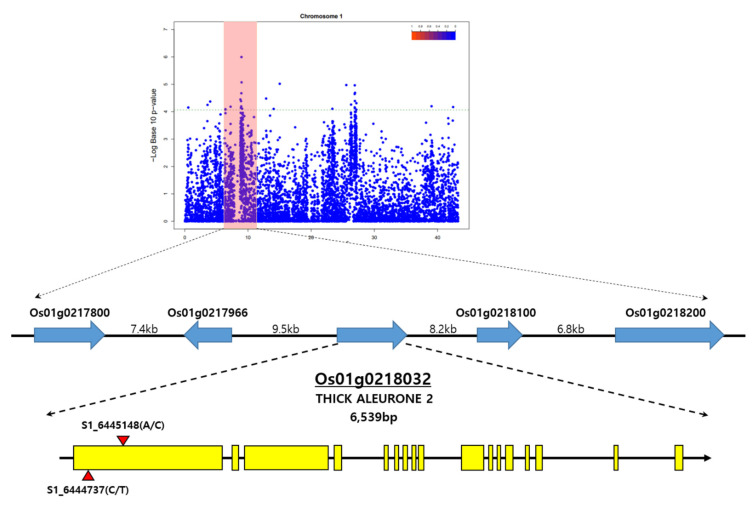
Manhattan plot derived from a GWAS scan for a traits of seed coat and aleurone layer thickness, thick aleurone 2 (TA2) gene structure, and its SNP location.

**Figure 7 genes-13-01805-f007:**
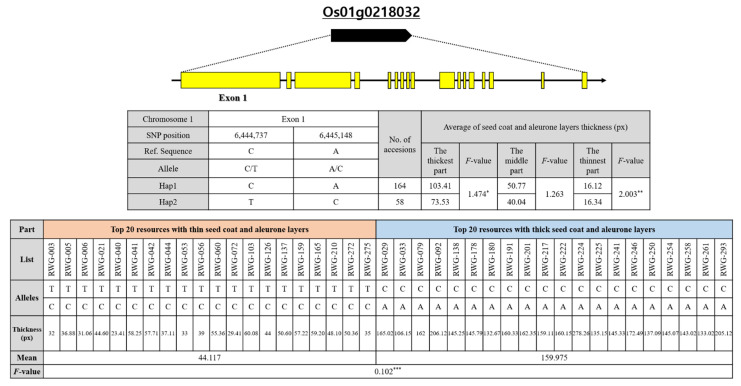
Haplotype analysis of the sequence covering the candidate gene *Os01g0218032* on chromosome 1 coding regions in the 294 rice core collection and the phenotypic variation among different haplotypes. *****, significant difference at *p* < 0.05; **, significant difference at *p* < 0.01; ***, significant difference at *p* < 0.001.

**Figure 8 genes-13-01805-f008:**
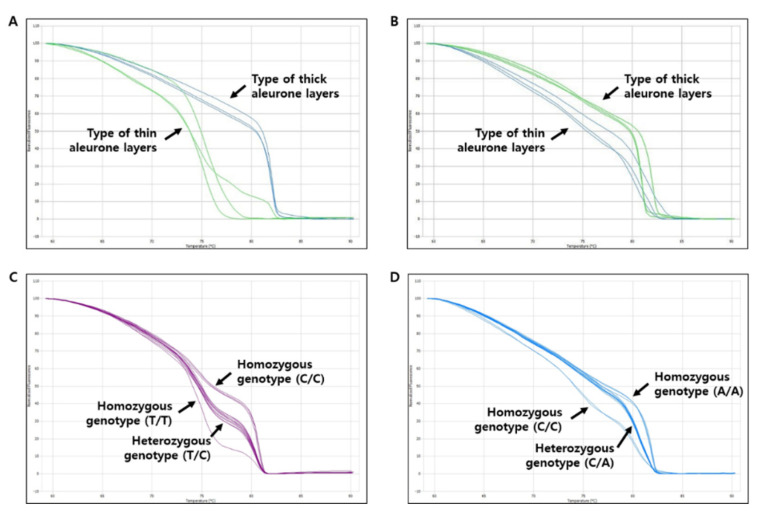
Validation test of the two HRM markers, TA2S1-737 (**A**,**C**) and TA2S1-148 (**B**,**D**), using the melting curves developed in this study. In A and B, three resources (RWG092, RWG224, and RWG292) with thick aleurone layers and three resources (RWG040, RWG053, and RWG072) with thin aleurone layers were used, respectively. In C and D, 50 individuals that crossed “Samgwang” (thick seed coat and aleurone layers) and “Seolgaeng” (thin seed coat and aleurone layers) were used, respectively.

**Table 1 genes-13-01805-t001:** Summary of statistics for thickness of seed coat and aleurone layers by region in the 294 rice genetic resources.

Parameter	Thickest Part of Aleurone Layers	Middle Part of Aleurone Layers	Thinnest Part of Aleurone Layers
Max	278.26	111.61	41.62
Min	6.57	18.44	4.30
Mean	102.95	50.76	15.59
SD *	33.58	15.90	6.20
CV	33.0	31.0	40.0
*F*-value	5.80 ***	5.41 ***	1.57 **

* SD, standard deviation; CV, coefficient of variation in percent (%); **, significant difference at *p* < 0.01; ***, significant difference at *p* < 0.001.

**Table 2 genes-13-01805-t002:** Single nucleotide polymorphism (SNPs) locations associated with the thickness of seed coat and aleurone layers in the 294 rice core collection.

Region of Aleurone Layers	SNP	Chr. *	Position	−log_10_(*p*)	MAF	R^2^
Thickest part	S4_34033252	4	34,033,252	5.94	0.37	0.27
	S5_22192204	5	22,192,204	5.63	0.29	0.27
	S5_13860270	5	13,860,270	5.45	0.25	0.26
	S10_9474845	10	9,474,845	5.39	0.34	0.26
	S5_576613	5	576,613	5.35	0.08	0.26
	S3_21376842	3	21,376,842	5.27	0.22	0.26
	S1_37703964	1	37,703,964	5.18	0.38	0.26
	S1_23339931	1	23,339,931	5.14	0.10	0.26
	S10_13747884	10	13,747,884	5.13	0.20	0.26
	S7_15563131	7	15,563,131	5.13	0.32	0.26
Middle part	S3_20312675	3	20,312,675	6.53	0.22	0.24
	S7_15626500	7	15,626,500	6.45	0.40	0.23
	S3_20316038	3	20,316,038	5.87	0.48	0.23
	S10_13717536	10	13,717,536	5.58	0.32	0.22
	S3_20362684	3	20,362,684	5.57	0.24	0.22
	S3_20375875	3	20,375,875	5.37	0.17	0.22
	S3_24286249	3	24,286,249	5.37	0.27	0.22
	S10_13483606	10	13,483,606	5.34	0.11	0.22
	S10_13633474	10	13,633,474	5.34	0.19	0.22
	S3_24392449	3	24,392,449	5.34	0.18	0.22
Thinnest part	S1_6459856	1	6,459,856	6.00	0.18	0.10
	S10_13753396	10	13,753,396	5.60	0.34	0.10
	S7_13503023	7	13,503,023	5.49	0.34	0.09
	S3_24411338	3	24,411,338	5.38	0.19	0.09
	S10_13677461	10	13,677,461	5.37	0.10	0.09
	S7_13769864	7	13,769,864	5.22	0.14	0.09
	S7_13614908	7	13,614,908	5.19	0.28	0.09
	S7_13769867	7	13,769,867	5.17	0.14	0.09
	S10_13481587	10	13,481,587	5.17	0.15	0.09
	S3_24326767	3	24,326,767	5.13	0.48	0.09

* Chr., chromosome; Position, position of SNP; MAF, minor allele frequency; R^2^, co-efficient of determination.

**Table 3 genes-13-01805-t003:** Information of genes identified in the genomic regions detected through the GWAS on the thickness of the seed coat and aleurone layers.

No.	SNP	Locus ID	Description	Position	Length (bp)	Gene Name
1	S1_6459856	Os01g0218032	Putative DNA demethylase, Endosperm development	chr01:6444246..6456068 (+ strand)	11,823	REPRESSOR OF SILENCING 1a, thick aleurone 2
2		Os01g0218100	Helix-loop-helix DNA-binding domain containing protein	chr01:6462770..6465035 (+ strand)	2266	-
3		Os01g0218200	Thioredoxin domain 2 containing protein	chr01:6471789..6476664 (+ strand)	4876	-
4		Os01g0218500	FLOWERING LOCUS T (FT)-Like homolog	chr01:6494446..6499766 (+ strand)	5321	FT-Like homolog
5	S1_23339931	Os01g0594300	Pistil-specific extensin-like protein family protein	chr01:23285238..23287336 (+ strand)	2099	-
6	S1_37703964	Os01g0850100	Similar to Phosphatidic acid phosphatase-like protein	chr01:36540164..36545196 (+ strand)	5033	-
7		Os01g0851300	Reticulon family protein	chr01:36652530..36654649 (− strand)	2120	-
8		Os01g0851600	3-oxo-5-α-steroid 4-dehydrogenase, C-terminal domain containing protein	chr01:36666965..36668075 (+ strand)	1111	-
9		Os01g0851700	Similar to Cytosolic starch phosphorylase (Fragment)	chr01:36670321..36676478 (− strand)	6158	-
10		Os01g0852200	Similar to sialin	chr01:36687545..36695629 (− strand)	8085	-
11		Os01g0852650	FAR1 DNA binding domain domain containing protein	chr01:36727945..36729912 (+ strand)	1968	-
12		Os01g0853400	Component of the SCF E3 ubiquitin ligase complex, Jasmonate-regulated defense responses, Promoting leaf senescence	chr01:36747521..36750925 (+ strand)	3405	CORONATINE INSENSITIVE1
13	S3_20312675	Os03g0564200	Protein of unknown function DUF952 family protein	chr03:20313408..20318292 (+ strand)	727	-
14		Os03g0564350	Conserved hypothetical protein.	chr03:20321430..20321672 (+ strand)	243	-
15	S3_20375875	Os03g0565100	OST3/OST6 family protein	chr03:20363579..20370035 (− strand)	6457	-
16		Os03g0565200	Haloacid dehalogenase-like hydrolase domain containing protein	chr03:20379625..20387042 (+ strand)	7418	-
17	S3_21376842	Os03g0565500	Similar to Elongation factor G 1, mitochondrial precursor (mEF-G-1) (EFGM)	chr03:20403051..20417506 (+ strand)	14,456	-
18		Os03g0565600	Similar to tobamovirus multiplication-like protein	chr03:20420795..20426307 (− strand)	5513	-
19		Os03g0581600	Similar to DAG protein	chr03:21363967..21368868 (+ strand)	4902	-
20	S3_24286249	Os03g0635100	Heterotrimeric G protein γ subunit 1, Regulation of abiotic stresses, Salinity stress tolerance	chr03:24252686..24256939 (− strand)	4,254	Heterotrimeric G-protein γ subunit 1
21	S3_24326767	Os03g0636800	LSTK-1-like kinase	chr03:24351648..24357284 (− strand)	5637	-
22	S3_24392449	Os03g0637600	Leucine-rich repeat, plant specific containing protein	chr03:24401251..24402892 (− strand)	1642	-
23		Os03g0637800	Crinkly4 receptor-like kinase, Epidermal cell differentiation, Interlocking of the palea and lemma, Control of grain size and shape	chr03:24415884..24420065 (+ strand)	4182	CRINKLY4
24	S3_24411338	Os03g0638200	Similar to Major facilitator superfamily protein, expressed	chr03:24449135..24453890 (+ strand)	4756	-
25		Os03g0638300	Tesmin/TSO1-like, CXC domain containing protein	chr03:24454477..24457481 (− strand)	3005	-
26	S4_34033252	Os04g0666500	Indole-3-acetate O-methyltransferase 1-like	chr04:34022617..34024079 (+ strand)	1167	-
27		Os04g0666800	Dirigent protein 2	chr04:34038375..34039430 (− strand)	773	-
28		Os04g0666900	Similar to H1005F08.22 protein	chr04:34040123..34048308 (− strand)	8186	-
29		Os04g0667200	Similar to H1005F08.24 protein	chr04:34060040..34063474 (+ strand)	3435	-
30		Os04g0667400	Melatonin 2-hydroxylase, Control of the melatonin level in plants	chr04:34067654..34070800 (− strand)	3147	Melatonin 2-hydroxylase
31	S5_576613	Os05g0110000	Zinc finger, RING/FYVE/PHD-type domain containing protein	chr05:538304..539630 (+ strand)	1327	-
32		Os05g0110200	DVL family protein	chr05:548370..548715 (− strand)	346	-
33		Os05g0110700	Chromosome segregation protein Spc25 domain containing protein	chr05:568790..572502 (− strand)	3713	-
34		Os05g0110900	Receptor-like cytoplasmic kinase, Positive regulation of peptidoglycan and chitin triggered immunity	chr05:577182..580396 (− strand)	3215	Receptor-like cytoplasmic kinase 176
35		Os05g0111000	Similar to Gag polyprotein	chr05:593356..597828 (+ strand)	4473	-
36		Os05g0111100	Zinc finger, Tim10/DDP-type family protein	chr05:599130..599776 (− strand)	647	-
37	S5_13860270	Os05g0306000	GOLD domain containing protein	chr05:13840559..13848909 (− strand)	1762	-
38	S5_22192204	Os05g0451100	Major facilitator superfamily protein	chr05:22155979..22159703 (+ strand)	3725	-
39		Os05g0451900	O-fucosyltransferase 19	chr05:22184180..22188759 (+ strand)	2171	-
40		Os05g0452400	Conserved hypothetical protein	chr05:22192448..22192819 (+ strand)	372	-
41		Os05g0452600	Similar to 50S ribosomal protein L33	chr05:22198315..22199542 (+ strand)	1228	-
42		Os05g0452900	Similar to phosphatidic acid phosphatase-related/PAP2-related	chr05:22214698..22219617 (− strand)	4920	-
43	S7_13503023	Os07g0420700	Similar to α-glucosidase like protein	chr07:13488360..13501017 (+ strand)	12,658	-
44		Os07g0420900	F-box protein 368	chr07:13509977..13510777 (+ strand)	801	-
45		Os07g0421000	Cyclin-like F-box domain containing protein	chr07:13513636..13515776 (− strand)	1530	-
46		Os07g0421300	Similar to α glucosidase-like protein	chr07:13534771..13548698 (+ strand)	13,928	-
47	S7_13614908	Os07g0421600	Similar to transferase family protein	chr07:13556183..13558261 (+ strand)	2079	-
48		Os07g0423000	Mitochodrial transcription termination factor-related family protein	chr07:13654341..13655841 (− strand)	1501	-
49	S7_13769864	Os07g0424400	Similar to Cellulose synthase-7	chr07:13741551..13747205 (− strand)	5655	-
50		Os07g0425000	Biopterin transport-related protein BT1 family protein	chr07:13776950..13778155 (− strand)	1206	-
51	S7_15563131	Os07g0451300	Cytochrome P450 family protein	chr07:15560341..15561945 (+ strand)	1605	-
52		Os07g0452100	Similar to α-galactosidase	chr07:15580083..15584254 (+ strand)	4172	-
53		Os10g0399200	Similar to Cys/Met metabolism PLP-dependent enzyme family protein	chr10:13449222..13450950 (− strand)	1729	-
54	S10_13481587	Os10g0399700	Similar to Cys/Met metabolism PLP-dependent enzyme family protein	chr10:13467597..13469366 (+ strand)	1770	-
55		Os10g0400100	Methionyl-tRNA synthetase, class Ia domain containing protein	chr10:13492802..13496772 (+ strand)	3971	-
56	S10_13483606	Os10g0400500	Similar to Pyridoxal-dependent decarboxylase conserved domain containing protein	chr10:13524417..13525925 (− strand)	1509	-
57	S10_13633474	Os10g0402200	Origin recognition complex subunit 3, Lateral root development	chr10:13600900..13606431 (− strand)	5532	ORIGIN RECOGNITION COMPLEX 3
58		Os10g0402400	Transferase family protein	chr10:13611824..13614202 (− strand)	2379	-
59	S10_13717536	Os10g0403000	Cytochrome P450 protein, CYP78A11, Regulation of leaf initiation rate and vegetative-reproductive phase change	chr10:13658790..13660543 (− strand)	1754	PLASTOCHRON 1
60		Os10g0403800	Basic helix-loop-helix (bHLH) transcriptional factor, Regulation of leaf angle	chr10:13721970..13722965 (− strand)	996	Basic helix-loop-helix protein 174
61	S10_13753396	Os10g0404000	Conserved hypothetical protein	chr10:13757771..13762009 (+ strand)	2046	-

**Table 4 genes-13-01805-t004:** Information of the two SNP markers for selecting the seed coat and aleurone layers as HRM PCR markers.

Gene	SNP Position (bp)	Allele	Oligo Name	Sequence (5′→3′)	Tm (°C)	Product Size
TA2	S1_6444737	C/T	TA2S1-737-Fw	CACGGAAACCCAAGAAGAAA	60	175 bp
			TA2S1-737-Rv	GCCTGTTCTGCAGGAGGTT	60	
	S1_6445148	A/C	TA2S1-148-Fw	TGCACATTTGTTTCCTCCTG	60	225 bp
			TA2S1-148-Rv	TGCGTCTGACTGATTGAACTG	60	

**Table 5 genes-13-01805-t005:** Statistical analysis of the HRM results with backcross population (BC_2_F_2_) from the cross between Samgwang (variety with a thick aleurone layers) and Seolgaeng (variety with a thin aleurone layers).

Population	Generation	Marker	Number of Plant				χ^2^ *	*p*-Value
			Total	Homo (Samgwang)	Hetero	Homo (Seolgaeng)		
Samgwang X Seolgaeng	BC_2_F_2_	TA2S1-737	50	12	30	8	2.64	0.27
		TA2S1-148	50	12	26	12	0.08	0.96

* d.f = 2: χ2 (0.05, 2) = 5.99.

## Data Availability

The datasets analyzed during the current study are available from the corresponding author upon reasonable request.

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
