# Peer review of "Development of SNP Markers from GWAS for Selecting Seed Coat and Aleurone Layers in Brown Rice (Oryza sativa L.)"

_genes, 2022, doi:10.3390/genes13101805_

Round 1
Reviewer 1 Report
This study used 294 rice core collection to investigate the thickness of seed coat and aleurone layers. Candidate gene related to thickness of seed coat and aleurone layers was reported by performing GWAS using SNPs throughout whole genome re-sequencing.
The paper in general is of interest, however, there should be a careful revision and write-up for all sections of the paper.
All comments have been made directly into the MS (attached).

Author Response
Response to Reviewer 1
We appreciate the comments that the reviewers have given in our manuscript and the constructive criticism the reviewer has given. We have carefully reviewed the comments and have revised the manuscript accordingly. We believe that these changes have clearly improved our manuscript.
Line 65 : You need to rewrite this paragraph just to lead you towards the importance of the core collection in studies related to yours
--- Thank you for the critical comments. We have described the objectives in Lines 62 as follows: The reduced size of core collection is key to manageability and represents diversity of entire collection, allowing analysis to be handled more efficiently and effectively.
Line 276 : I think the whole discussion section needs to be revised carefully. Please give your key findings and discuss them against previous related research and emphasize the novelty of your findings in the proper context. Please also give proper interpretations to your results including any possible limitations
--- Thank you for the critical comments. We have carefully described the discussion part through core results in Discussion section.
Line 277 : You are not discussing the GWAS. Instead please discuss your results
--- Thank you for the critical comments. We have described the objectives in Lines 359~365 as follows: According to genome resequencing results of 294 rice core collection, the number of sequence reads was about 42 million, and mapping rate was over 96 %. The average coverage was 9.8 depth, and the total number of SNPs was 11,632,676. In core collection, the number of SNPs for 30 landrace species was 7,239,907, and the number of SNPs was 6,792,796 for 30 weedy rice. The number of SNPs was 10,728,014 with 65 introduced lines. These results support the fact that core collection used in this study is suitable for GWAS.
Line 284 : Is this is really a discussion related to your results?. This is more of introduction nature
--- Thank you for the critical comments. We have described the objectives in Lines 376~390 as follows: GWAS was performed using phenotypic data on seed coat and aleurone layers thickness of 294 rice resources in this study. It was confirmed that significant SNPs were detected based on –log10(p) > 5 in Manhattan plot according to thickness of seed coat and aleurone layers. In addition, expected P-value of x-axis and observed P-value from significant SNPs showed a diagonal linear shape in QQ plot, which means that the discovered SNPs express their characteristics well with normality and significance. In results of LD block analysis based on SNP with the highest correlation according to thickness of seed coat and aleurone layers, various LD blocks were observed on chromosome 1, 3, and 4, among them, S1_6459856 was determined to be a significant SNP, and ROS1 was selected as the closest candidate gene. As a result of haplotype analysis with Os01g0218032 on basis of the high peak position of rice chromosome 1, Os01g0218032 gene was divided into two groups. Os01g0218032-Hap1 group showing a thick seed coat and aleurone layers had a base similar to Nipponbare in exon region, whereas Os01g0218032-Hap2 group showing a thin seed coat and aleurone layers thickness had a base substitution at exon region.
Line 289 : Again please link any previous study or report to your own work. These are general statements
--- Thank you for the critical comments. We have described the objectives in Lines 376~390.
-----------------------------------------------------------
--- Thank you for the critical comments. In addition, we have revised all minor points in the text of the manuscript.

Reviewer 2 Report
This manuscript described a GWAS study for the thickness of rice seed coat and aleurone layers, and a candidate gene was identified. Then, two primer pairs of HRM markers were developed, and validated using a segregating population. The result has high value for breeding varieties with high nutrition. However, the results have some problems to solve.
(1) the threshold of GWAS was described as –log>8 in section 2.4, why did it change to –log>5 in section 3.2?
(2) How long is the LD distance in these germplasm?
(3) In section 3.2, the authors only showed top 10 significant SNPs, why not showed all significant SNPs? If they were too many, authors can show the most significant SNPs in a LD distance.
(4) In section 3.3, why did author identify candidate genes within the range of ±5 kb? They should identify all annotated genes within LD distance.
(5) In section 3.3, authors should add the result of haplotype analysis of Os01g0218021 using the two SNP C6444737T and A6445148C to confirm that this gene is genetically true for the thickness of rice seed coat and aleurone layers.
(6) In section 3.4, authors should add the results whether the thicknesses of rice seed coat and aleurone layers of three genotypes have significant difference.
(7) In section 3.4, 30 and 8 individuals were heterozygous and Seolgaeng homozygous genotypes by TA2S1-737, while 26 and 12 individuals were heterozygous and Seolgaeng homozygous genotypes by TA2S1-148. These two markers were both located in Os01g0218021, why did so much different between them? Did they have so much recombinant?
Minor problem:
The title of “3.3. Development of molecular markers for traits of seed coat and aleurone layers thickness” should be corrected to “3.4. Development of molecular markers for traits of seed coat and aleurone layers thickness”.
Author Response
Response to Reviewer 2
We appreciate the comments that the reviewers have given in our manuscript and the constructive criticism the reviewer has given. We have carefully reviewed the comments and have revised the manuscript accordingly. We believe that these changes have clearly improved our manuscript.
This manuscript described a GWAS study for the thickness of rice seed coat and aleurone layers, and a candidate gene was identified. Then, two primer pairs of HRM markers were developed, and validated using a segregating population. The result has high value for breeding varieties with high nutrition. However, the results have some problems to solve.
(1) the threshold of GWAS was described as –log>8 in section 2.4, why did it change to –log>5 in section 3.2?
--- Thank you for the critical comments. We have described the objectives in Lines 125 as follows: Significant SNPs were determined based on the threshold of – log P > 5.
(2) How long is the LD distance in these germplasm?
--- Thank you for the critical comments. We have revised the comments in Lines 126~136, “Materials and Methods, 2.5” & in Lines 240~242 as follows:
2.5. Analysis of LD block and haplotype
We used Haploview (v.4.2) for analysis of LD block and haplotype analysis [47]. We defined LD blocks according to definition of haplotype block described by Gabriel et al. [48]. This method defines pairs to be in strong LD if the one-sided upper 95% confidence bound on D′ is >0.98 and the lower bound is above 0.7. Each color represents the magnitude and significance of pairs LD, and the gradation from red to white is from a high LD value to a low value. Red diamonds without numbers correspond to a D' value of 1.0. LD analysis was used to designate a region containing 10 kb either side of lead SNP as the candidate region. Annotation of genes within candidate regions were derived from the Rice Annotation Project database (https://rapdb.dna.affrc.go.jp). Haplotypes were used for comparative phenotypic analysis.
LD block analysis was performed at 10 kb region between data identified as valid SNPs using Haploview (v.4.2).
(3) In section 3.2, the authors only showed top 10 significant SNPs, why not showed all significant SNPs? If they were too many, authors can show the most significant SNPs in a LD distance.
--- Thank you for the critical comments. As a result of GWAS analysis, the number of SNPs identified in peak region was detected so many that it was difficult to provide all information. To replace this point, LD analysis was performed to prove association with the top 10 SNPs we selected. We have revised it in Lines 234~246 according to the comments.
To give more information, we have added the results of PCA and kinship analyses in Lines 197~209 as follows: PCA and kinship analyses were performed for population structure analysis using high-quality 1,842,515 SNPs with a compressed mixed linear model (CMLM). 294 rice core collection panel was determined by graphical representation of K value on scree plot, which indicated the presence of three sub-populations. This result was confirmed by variation in the first 10 major components (PCs) showed an inflection points in PC2, suggesting that the first two PCs dominated the population structure on the association analysis (Figure 3). The first two PCs were significant and explained 39.65%, 3.7% of total genetic variation which suggests the predominance of three components in controlling population structure of association panel. Kinship matrix is displayed as a heatmap, where red indicates the highest correlation between pairs of individuals and yellow indicated the lowest correlation. The relatedness of pairwise coefficients estimated in the kinship matrix indicated the lower genetic relatedness among individuals in the association panel.
(4) In section 3.3, why did author identify candidate genes within the range of ±5 kb? They should identify all annotated genes within LD distance.
--- Thank you for the critical comments. We performed LD analysis on 10 kb region based on the significant SNP. Among them, genes whose functions were specified and reported for association with the thickness of seed coat and aleurone layers were selected. We have described these in Lines 234~246.
(5) In section 3.3, authors should add the result of haplotype analysis of Os01g0218021 using the two SNP C6444737T and A6445148C to confirm that this gene is genetically true for the thickness of rice seed coat and aleurone layers.
--- Thank you for the critical comments. We have described the objectives in Lines 280~296 as follows: In Exon1 region of OsROS1, a total of six SNPs closed to S1_6459856 were formed LD blocks in same block as S1_6459856. This is a result of informing that there is an LD rela-tionship between SNPs. Among the SNP markers in which LD block was formed, haplo-type for each individual was estimated using Haploview for S1_6444737 and S1_6445148 (Figure 7). For haplotype analysis, Nipponbare nucleotide sequence was used as standard. Haplotype analysis results and phenotypic data were compared and analyzed to deter-mine whether there was a relationship between the thickness of seed coat and aleurone layers and the candidate gene. As a result of haplotype analysis with Os01g0218032 on basis of the high peak position of rice chromosome 1, Os01g0218032 gene was divided into two groups, and the average thickness of each group was 103.41/73.53 in the thickest part, 50.77/40.04 in the middle part, and 16.12/16.34 in the thinnest part. Os01g0218032-Hap1 group showing a thick seed coat and aleurone layers had a base similar to Nipponbare in exon region, whereas Os01g0218032-Hap2 group showing a thin seed coat and aleurone layers thickness had a base substitution at exon region.
(6) In section 3.4, authors should add the results whether the thicknesses of rice seed coat and aleurone layers of three genotypes have significant difference.
--- Thank you for the critical comments. This is described in Lines 291~293 as follows: There was a significant difference among three genotypes of the thicknesses (with averages of 103.41, 50.77, & 16.12 px) of rice seed coat and aleurone layers (data not shown).
(7) In section 3.4, 30 and 8 individuals were heterozygous and Seolgaeng homozygous genotypes by TA2S1-737, while 26 and 12 individuals were heterozygous and Seolgaeng homozygous genotypes by TA2S1-148. These two markers were both located in Os01g0218021, why did so much different between them? Did they have so much recombinant?
--- Thank you for the critical comments. Both markers are located in Os01g0218032 region, but as a result of haplotype analysis, it can be seen that genotype different from reference sequence affects the thin seed coat and aleurone layers thickness. We confirmed that melting curve was sufficiently differentiated according to two alleles (C/T, A/C combination) when making HRM primers.
Minor problem:
The title of “3.3. Development of molecular markers for traits of seed coat and aleurone layers thickness” should be corrected to “3.4. Development of molecular markers for traits of seed coat and aleurone layers thickness”.
--- Thank you for the comments. We have described this in Lines 309.

Round 2
Reviewer 1 Report
Dear authors,
Although your research area is interesting, I think the way you are presenting your results is greatly affecting the quality of your paper. I am not convinced with what you have done in the first round of the review especially in the "Results" and "Discussion" sections. So, please carefully go through your MS and try to improve your write-up of your results, the discussion and the language. I have made some comments directly in the text of the MS (see the attached file).

Author Response
Response to Reviewer 1
- I think you need to carefully go through your results write-up:
(1) avoid repeating the materials and methodologies again here unless it is essential.
(2) present your results in a clear and understandable way and avoid long sentences
(3) check the language especially the newly added sentences
---Thank you for your critical comments. To provide a basic information for readers, we need to give a little background of the analysis in the Results. We revised all parts according to the comments
- I am here repeating what I have mentioned in the 1st review round. I think the authors did very little to improve the "Discussion" section. {I think the whole discussion section needs to be revised carefully. Please give your key findings and discuss them against previous related research and emphasize the novelty of your findings in the proper context. Please also give proper interpretations to your results including any possible limitations. }
---Thank you for your critical comments. We have carefully revised 'Discussion' section in Lines 342~438 as follow: (1) Accuracy of seed coat and aleurone layers staining and measurement method, (2) GWAS using high-quality resequencing data of 294 rice core collection, (3) Discovery of candidate genes related to thickness of seed coat and aleurone layers, (4) Polymorphism test of HRM markers developed in this study for cross population.
- Let me summarize your key findings:
(1) Significant phenotypic variations in thickness of seed coat and aleurone layers were found
(2) SNPs correlated with seed coat and aleuronat layers’ thickness and were discovered
(3) Candidate gene related to thickness of seed coat and aleurone layers were identified.
(4) Two primer pairs that can be used as high-resolution melting (HRM) markers were developed for selecting seed coat and aleurone layer thickness.
I think most of your discussion should focus on these results.
---Thank you for your critical comments. We have carefully revised 'Discussion' section as follow: (1) Accuracy of seed coat and aleurone layers staining and measurement method, (2) GWAS using high-quality resequencing data of 294 rice core collection, (3) Discovery of candidate genes related to thickness of seed coat and aleurone layers, (4) Polymorphism test of HRM markers developed in this study for cross population. We described as follows in Lines 342, 374, 386, and 429.
- I do not think it is a good idea to start the "Discussion" section in this way. Please start with your key findings to attract the readers from the beginning instead of starting with a whole paragraph to show the usefulness of GWAS.
---Thank you for your critical comments. We revised the ‘Discussion” in Line 342 as follows: In rice grains, embryo and aleurone layers are major tissues in which lipids are accumulated [27], lipid component of brown rice is not only an essential energy source for plant germination and growth, but also an important nutrient source for humans. However, when polished, most of these useful substances are removed, and white rice has the disadvantage of losing food-related ingredients, such as umami-related amino acids, when compared to the unpolished group.
- This has been repeated several times. I think you do not need to do so all the time.
---Thank you for your critical comments. We revised it in Line 386 as follows: In this study, GWAS was performed using high-quality resequencing data from 294 rice core collection and phenotypic data using a staining method with verified significance.
- Minor comments
---All minor comments were well revised.

Reviewer 2 Report
The revised MS has improved much. However, there are still two questions to be solved:
1. As we know, the average LD distance of rice is about 200 kb, and even 500 kb in japonica rice, why did the author perform the LD analysis at 10 kb region.
2. The 50 BC2F2 population was derived from Samgwang and Seolgaeng. This means there were two haplotypes in the population. TA2S1-373 and TA2S1-148 were both located in the candidate gene Os01g0218032, and all individuals should be the same genotypes by both of them, except recombination between these two site or mistake of genotyping. 30 and 8 individuals were heterozygous and Seolgaeng homozygous genotypes by TA2S1-737, while 26 and 12 individuals were heterozygous and Seolgaeng homozygous genotypes by TA2S1-148. Authors should check the problem again.
Author Response
Response to Reviewer 2
The revised MS has improved much. However, there are still two questions to be solved:
- As we know, the average LD distance of rice is about 200 kb, and even 500 kb in japonica rice, why did the author perform the LD analysis at 10 kb region.
----Thank you for your critical comments. In case of LD block, it is very flexible depending on characteristics of group, and location or size of LD block shows a very large difference. As described below, elite breeding line in which a gene is fixed has a high LD, but landrace used in this study shows a low LD. We described it in Lines 222~224 as follows: LD decay ranges from less than 1 kb in landraces to over 100 kb in elite breeding lines. In other words, higher LDs in better elite lines and lower LDs in more differentiated or foreign varieties are shown (Vos et al., 2016; Ching et al., 2002; Yan et al., 2010).
- The 50 BC2F2 population was derived from Samgwang and Seolgaeng. This means there were two haplotypes in the population. TA2S1-373 and TA2S1-148 were both located in the candidate gene Os01g0218032, and all individuals should be the same genotypes by both of them, except recombination between these two site or mistake of genotyping. 30 and 8 individuals were heterozygous and Seolgaeng homozygous genotypes by TA2S1-737, while 26 and 12 individuals were heterozygous and Seolgaeng homozygous genotypes by TA2S1-148. Authors should check the problem again.
----Thank you for your critical comments. According to several HRM marker analysis researches, it is possible to develop multiple HRM markers targeting SNPs within one gene. In addition, these markers were sufficiently correlated with target phenotype in the mutant, crossed line, and variety (Ishikawa et al., 2010; Chandra et al., 2022). In this study, a chi-square test was performed according to generation separation ratio using two developed HRM markers. As a result, since χ2 (our result values; 2.64 and 0.08) is smaller than χ2 (0.05,2) 5.99, it is judged that it has an effective result on HRM marker and generation separation ratio.
(Reference)
Ishikawa, T., Kamei, Y., Otozai, S., Kim, J., Sato, A., Kuwahara, Y., Tanaka, M., Deguchi, T., Inohara, H., Tsujimura, T., & Todo, T. (2010). High-resolution melting curve analysis for rapid detection of mutations in a Medaka TILLING library. BMC Molecular Biology, 11, 70 - 70.
Chandra, S., Oh, Y., Han, H., Salinas, N., Anciro, A.L., Whitaker, V.M., Chacón, J.G., Fernandez, G.E., & Lee, S. (2021). Comparative Transcriptome Analysis to Identify Candidate Genes for FaRCg1 Conferring Resistance Against Colletotrichum gloeosporioides in Cultivated Strawberry (Fragaria × ananassa). Frontiers in Genetics, 12.

Round 3
Reviewer 1 Report
Thank you for your revision with which I believe the paper is now in a better shape. However, I think you can do more to further improve the quality of your paper. Some comments are in the attached file.

Author Response
Thank you very much for reviewer’s detailed comments. We were able to improve our manuscript very well.
- Line 160, incomplete sentence
--- corrected
- Do you think the CV is used to measure the genetic variation or rather the precision of your experiment?
--- The following senctence is deleted as in Lines 161~163 : This statistics values suggested that there were abundant genetic variations in seed coat and aleurone layers thickness, and significant phenotypic variations were observed for 294 rice resources.
- Always avoid starting a new sentence in this way. Instead you can simply say The LD....
If this is a result of your study then should be mentioned in the past tense. The LD decay ranged from.....
--- The sentence is revised according to the comments.
- Please change variety with another word, e.g. type
--- We have revised all points according to the comments.

Reviewer 2 Report
Although the author have revised the MS two times. I am sorry that I still could not agree to publish with two reasons. Firstly, the candiate gene analysis of GWAS study is largely dependenting on LD. In Rice, the average genomic LD distance is about 200 kb. Although the author showed LD decay ranged from 1 kb to 100 kb, all three literatures were not rice. The authorS screened only 10 kb region around the most significant SNP, many candidate gene informations were lost. Secondly, the genotyping using both two HRM markers showed Medelian inheritance of ROS1a. However, the genotypes of 50 individuals by the two markers were inconsistent. This result showed there were some mistakes of genotyping by these two makers and would largely affect their values of application in molecular breeding.
Although there are some problems in the study, it's wtill a vary important work for enhancing the nutrients in rice grains. Thus, I encourage the authors reanalyze their data and then resubmit.
Author Response
- Although the author have revised the MS two times. I am sorry that I still could not agree to publish with two reasons. Firstly, the candidate gene analysis of GWAS study is largely depending on LD. In Rice, the average genomic LD distance is about 200 kb. Although the author showed LD decay ranged from 1 kb to 100 kb, all three literatures were not rice. The authors screened only 10 kb region around the most significant SNP, many candidate gene informations were lost.
--- Authors have reanalyzed the wider region for 200 kb according to reviewer’s comments and have revised the part with the results in Lines 220~222 as follows: “LD block analysis was performed in both sides of 100 kb region on the location identified as valid SNPs using Haploview (v.4.2). The LD decay ranges from less than 1 kb in landraces to over 100 kb in elite breeding lines.”
* Authors have added the “Supplementary Figure 1” for the genes detected from the wider region for 200 kb according to the reviewer’s comments.
Revised in Lines 253~264 as follows: “In order to find candidate genes affecting seed coat and aleurone layers thickness, candidate genes were identified within the range of ±100 kb using RAP database (https://rapdb.dna.affrc.go.jp/index.html) and NCBI database (https://www.ncbi.nlm.nih.gov/gene/) based on the most significant SNP positions on each chromosome (Table 3, Figure S1). A total of 61 genes were identified on rice chromosomes 1, 3, 4, 5, 7, and 10, and detailed information about several genes was not reported and annotated as conserved hypothetical proteins. Except for the conserved hypothetical protein, genes related to floral signal such as FLOWERING LOCUS T (FT)-Like homolog, genes related to plant growth such as ORIGIN RECOGNITION COMPLEX 3, and genes related to stress resistance such as CORONATINE INSENSITIVE1 were identified. In addition, genes related to seed development, such as CRINKLY4 and REPRESSOR OF SILENCING 1a, were also identified in strong LD.”
- Secondly, the genotyping using both two HRM markers showed Medelian inheritance of ROS1a. However, the genotypes of 50 individuals by the two markers were inconsistent. This result showed there were some mistakes of genotyping by these two makers and would largely affect their values of application in molecular breeding.
---- Reviewer mentioned that “the genotyping using both two HRM markers showed Medelian inheritance of ROS1a. However, the genotypes of 50 individuals by the two markers were inconsistent.”
In this regard, we think the reviewer need to consider the reason of statistical analysis. The statistical analysis is carried out for the reason why we have to get the significance of the experimental data for Mendelian inheritance with many individuals, because the genetic segregation of the seed coat and aleurone layers in individual plants may not be coincided exactly with HRM genotypes.
- Although there are some problems in the study, it's a very important work for enhancing the nutrients in rice grains. Thus, I encourage the authors reanalyze their data and then resubmit.
---- Thank you for reviewer’s kind comments.
Response to Reviewer 2
- Although the author have revised the MS two times. I am sorry that I still could not agree to publish with two reasons. Firstly, the candidate gene analysis of GWAS study is largely depending on LD. In Rice, the average genomic LD distance is about 200 kb. Although the author showed LD decay ranged from 1 kb to 100 kb, all three literatures were not rice. The authors screened only 10 kb region around the most significant SNP, many candidate gene informations were lost.
* Authors have added the “Supplementary Figure 1” for the genes detected from the wider region for 200 kb according to the reviewer’s comments
--- According to the reviewer’s comment, candidate genes were searched for 200kb region based on lead SNP, and 61 candidate genes were detected for in this region and included in paper (Table 3, Supplementary Figure 1.).
* We have revised it in Materials and Methods as in Lines 112, and 125~126 as follows:
(Line 112, 208) the threshold of – log P > 5.
(Lines 125~126) Adjacent significant SNPs associated with target traits within a physical distance of 200 kb were screened for candidate genes.
* We deleted Lines 219~223, because it is not necessary here.
“The deleted part is as follows: LD block analysis was performed in both sides of 100 kb region on the location identified as valid SNPs using Haploview (v.4.2). The LD decay ranges from less than 1 kb in landraces to over 100 kb in elite breeding lines. In other words, higher LDs in better elite lines and lower LDs in more differentiated or foreign varieties are shown [47,48,49].
* We have removed three references (47,48,49) from the Reference that deleted in Lines 219~223.
* We also replaced Figure 4 based on the threshold of – log P > 5 in Lines 232~233.
– Figure 4 was replaced.
* We have revised it based on the reanalysis of 200 kb region according the reviewer’s comment in Lines 248~259 as follows:
“In order to find candidate genes affecting seed coat and aleurone layers thickness, candidate genes were identified within the range of 200 kb using RAP database (https://rapdb.dna.affrc.go.jp/index.html) and NCBI database (https://www.ncbi.nlm.nih.gov/gene/) based on the most significant SNP positions on each chromosome (Table 3, Figure S1). A total of 61 genes were identified on rice chromosomes 1, 3, 4, 5, 7, and 10, and detailed information about several genes was not reported and annotated as conserved hypothetical proteins. Except for the conserved hypothetical protein, genes related to floral signal such as FLOWERING LOCUS T (FT)-Like homolog, genes related to plant growth such as ORIGIN RECOGNITION COMPLEX 3, and genes related to stress resistance such as CORONATINE INSENSITIVE1 were identified. In addition, genes related to seed development, such as CRINKLY4 and REPRESSOR OF SILENCING 1a, were also identified in strong LD.”
Round 4
Reviewer 2 Report
This revised MS has been largely improved, and I have no other question. I suggest it could be accepted.